# Breaking constraint of mammalian axial formulae

Gabriel M. Hauswirth [1,2,5], Victoria C. Garside[1,2,5], Lisa S. F. Wong[1,2], Heidi Bildsoe[1,2], Jan Manent[1,2], Yi-Cheng Chang[1,2], Christian M. Nefzger[2,3,4], Jaber Firas[2,3,4], Joseph Chen[2,3,4], Fernando J. Rossello[2,3,4], Jose M. Polo [2,3,4] & Edwina McGlinn [1,2✉]

The vertebral column of individual mammalian species often exhibits remarkable robustness in the number and identity of vertebral elements that form (known as axial formulae). The genetic mechanism(s) underlying this constraint however remain ill-defined. Here, we reveal the interplay of three regulatory pathways (Gdf11, miR-196 and Retinoic acid) is essential in constraining total vertebral number and regional axial identity in the mouse, from cervical through to tail vertebrae. All three pathways have differing control over *Hox* cluster expression, with heterochronic and quantitative changes found to parallel changes in axial identity. However, our work reveals an additional role for *Hox* genes in supporting axial elongation within the tail region, providing important support for an emerging view that mammalian Hox function is not limited to imparting positional identity as the mammalian body plan is laid down. More broadly, this work provides a molecular framework to inter-rogate mechanisms of evolutionary change and congenital anomalies of the vertebral column.

[1] EMBL Australia, Monash University, Clayton, VIC 3800, Australia. [2] Australian Regenerative Medicine Institute, Monash University, Clayton, VIC 3800, Australia. [3] Department of Anatomy and Developmental Biology, Monash University, Clayton, VIC, Australia. [4] Development and Stem Cells Program, Monash Biomedicine Discovery Institute, Clayton, VIC, Australia. [5] These authors contributed equally: Gabriel M. Hauswirth, Victoria C. Garside. ✉email: edwina.mcglinn@monash.edu

The mammalian vertebral column is comprised of serially-repeating vertebrae that, while individually-identifiable, are grouped based on morphological and functional similarity into cervical (C), thoracic (T), lumbar (L), sacral (S) and caudal (Ca) elements. Vertebrae arise from developmentally-transient structures called somites, that are in turn generated from a progenitor pool located in the posterior embryo through the coordinated processes of axial elongation and segmentation[1]. The genetic mechanisms dictating vertebral identity are well understood, and centre on the temporally-controlled activation of Hox cluster genes[2] within vertebral progenitors[3]. In contrast, uncovering the genetic mechanisms that control vertebral number of each axial region in a meristic rather than homeotic manner (as defined in Bateson, 1894[4]), and consequently total vertebral number (TVN) of a mammalian species, has proved incredibly challenging.

The changes that constitute diversity of mammalian axial formulae are not uniform along the anterior-posterior (A-P; head-to-tail) axis[5,6]. Cervical number is almost exclusively fixed at 7, a trait that has rarely changed over 200 million years of mammalian evolution, spanning back to the mammalian ancestor of the late Triassic[7]. Two rare exceptions to this rule, manatee and sloth, have sparked intense interest in the advantage and developmental basis of these break in constraint[8–10], with their genetic basis unresolved to date. Thoraco-lumbar (T-L) count, while less rigid, is still highly conserved at 19 or 20 across most mammals, though again with examples of relaxation in both absolute T-L number and intraspecies robustness seen in clades such as Xenarthra and Afrotheria[11,12]. Sacral count varies considerably across mammals and along with the extreme diversity seen in caudal/tail morphology reinforces a graded decrease in constraint along the A-P axis.

As mouse displays the likely ancestral mammalian presacral formulae[6,13], and Rodentia exhibits the lowest deviation from median vertebral counts of all mammals[12], it should serve as an excellent model with which to dissect genetic mechanisms constraining axial formulae. However to date, although many truncating mouse mutants exist, often being highly dysmorphic, the ability to increase TVN in a meristic not simply homeotic manner has been limited to perturbation of only two genetic pathways[14–16] (see Supp Information for note on Hox13 mouse mutant phenotypes). The largest of these two meristic increases was observed in, perhaps intuitively, the evolutionary more plastic tail region; ectopic Lin-28 was found to increase the number of tail elements by 5 while let-7 opposed this function[14,15]. Within the less variable presacral region, microRNA(miR)-196 activity has been shown to constrain thoracic number by two, constituting one homeotic and one meristic change[16]. The signalling molecule Gdf11 has a far greater role than miR-196 in constraining T-L count as seen in conditional or complete knockout mice[17,18], and partial loss of Cdx transcription factors can increase presacral number by 1[19], but each does so at the expense of the caudal skeleton. Whether more subtle manipulations such as timing of signal onset[20] or exact expression levels provide a genetic basis for divergent mammalian morphologies, and more broadly, the degree to which genetic compensation or pleiotropic effects[21,22] underlmay evolutionary constraint of axial formulae, remains unresolved.

Here, we show that in vitro modelling of axial progenitors identifies unexpected cooperation between transcriptional and post-transcriptional regulatory mechanisms that act to terminate presacral expression signatures. Applying this knowledge in vivo, we are able to experimentally prolong the natural process of axial elongation in the mouse and considerably increase total vertebral number within both presacral and caudal regions. We reveal a high level of redundancy and synergism between Gdf11, miR-196

and retinoic acid (RA) in the constraint of axial formulae, with Gdf11 and RA cooperativity impacting a surprisingly large extent of the vertebral column including the constraint of cervical number to 7. All regulatory mechanisms converge in their ability to control spatio-temporal Hox expression, a parameter that our data supports having a substantive role in axial elongation, at least in the tail region.

## Results

**Synergistic constraint of trunk Hox expression signatures**. To dissect mechanisms constraining regionally-restricted Hox expression, as a proxy for regional axial morphology, we initially focused on modelling the trunk-to-tail (T-to-T) transition in vitro since progression through this critical moment in body axis formation correlates with the switch from an expanding to a depleting and eventually exhausted progenitor pool in species as diverse as mouse and snake[23]. miR-196 has a non-redundant role in timing the T-to-T transition, and thus we generated induced pluripotent stem cells (iPSCs) from wild-type (WT) and miR-196a1$^{gfp/gfp}$;miR-196a2$^{gfp/gfp}$;miR-196b$^{-/-}$ (miR-196-TKO) isogenic mice using standard protocols[24] (Supp. Figure 1a). Both iPSC lines were differentiated to model the developmental kinetics of a posterior growth zone by sequential addition of FGF2 on Day (D) 0 and the Wnt pathway agonist CHIR99021 on D2 (Fig. 1a, based previous work[25–28]). miR-196-TKO iPSCs were able to generate axial progenitors normally, however, exhibited marked differences in the kinetics of Hox cluster progression when compared to WT (Fig. 1b). Trunk Hox genes Hoxb8 and Hoxc8 displayed an identical timing of activation for both genotypes, but as differentiation proceeded, their quantitative level and temporal persistence were significantly increased in miR-196-TKO cells (Fig. 1b), consistent with the presence of functional miR-196 binding site(s) within their 3′UTRs[16,29]. Addition of Gdf11 to culture conditions was able to suppress trunk Hox gene expression and induce posterior Hox activation in both genotypes as anticipated. However, in the absence of Gdf11, both Hoxb8 and Hoxc8 displayed an unrestrained upwards trajectory in miR-196-TKO cells but not in WT (Fig. 1b). This key result indicated that miR-196 and Gdf11 are not simply redundant in this context, but act in concert in shutting down a trunk Hox code, a parameter suggested to support axial elongation under certain circumstances[30].

**Gdf11 and miR-196 synergistically constrain TVN**. To determine whether the combined function of Gdf11 and miR-196 identified in vitro influences axial elongation in vivo, we interbred miR-196-TKO[16] and Gdf11$^{-/-}$[17] mice and characterised TVN and vertebral identity across the allelic deletion series (Fig. 2a–d, Supp. Table 1). Consistent with previous reports, complete loss of miR-196 alone resulted in an increase of 2 thoracic elements (C7, T15, L5, median TVN 62), while complete loss of Gdf11 alone resulted in a dramatically expanded presacral region (C7, T18, L8/9) that truncated soon after sacral elements formed. In contrast to the Gdf11$^{-/-}$ truncation phenotype, we found that Gdf11$^{+/-}$ embryos presented with an additional 1 thoracic and 2 caudal vertebrae (C7, T14, L6, median TVN 64), a surprising dose-dependent effect of Gdf11 on TVN that has not previously been appreciated. Crossing miR-196-TKO onto this Gdf11$^{+/-}$ background increased the number of thoracic vertebrae further, producing viable offspring with a total of 5 additional elements (C7, T16, L6, median TVN 66). Gdf11$^{-/-}$;miR-196-TKO compound mutant embryos demonstrated a remarkable increase in the number of presacral elements with an additional 9 thoracic and 4 lumbar elements compared to WT (C7, T22, L10; Fig. 2b), the greatest expansion of presacral number reported in mice to

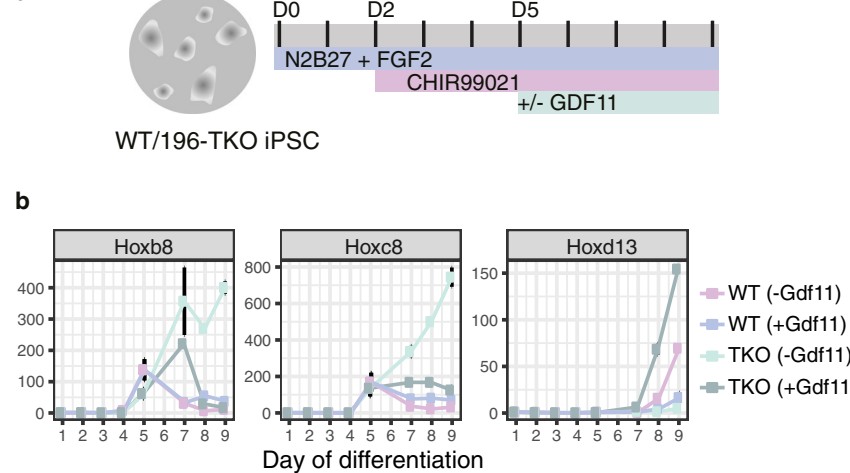

**Fig. 1 miR-196 and Gdf11 act synergistically in vitro to suppress a trunk *Hox* code. a** Schematic of 9-day in vitro differentiation protocol employed to model cell-state transitions in axial elongation and sequential *Hox* cluster activation. D = day. **b** Quantitative PCR analysis of trunk (*Hoxb8*, *Hoxc8*) and posterior (*Hoxd13*) *Hox* genes during in vitro differentiation of wild-type (WT) and *miR-196*-triple knockout (*miR-196-TKO*) induced pluripotent stem cells (iPSCs). Data points (coloured squares) represent the mean of technical triplicate and biological duplicate analysis (error bar = SEM).

date. Together, these data provide 3 major advances: (i) Gdf11 activity is required under wild-type conditions to constrain total vertebral number; (ii) The vertebral column is exquisitely sensitive to dosage of Gdf11, with opposing outcomes revealed dependent on allele number; (iii) miR-196 and Gdf11 do not constitute entirely parallel mechanisms in the control of presacral vertebrae number but rather, act synergistically, since the effect of their combined loss is greater than the sum of individual mutant phenotypes (Fig. 2d).

**Retinoic acid has pleiotropic effects on axial formulae.** While TVN cannot be accurately quantified in *Gdf11*[−/−] embryos due to the highly dysmorphic nature of post-sacral elements, the loss of miR-196 activity did not qualitatively appear to rescue *Gdf11*[−/−] truncation (Fig. 2a). In this context, the partial rescue of *Gdf11*[−/−] truncation by reducing endogenous levels of Retinoic acid (RA) signalling[31] led us to examine the potential for more elaborate redundancy or synergy between the three signalling/regulatory pathways. Indeed, oral gavage of pregnant dams across the genetic deletion series with pan-RA receptor inhibitor AGN193109 (AGN) revealed further combinatorial changes in vertebral number and identity, spanning each major subdivision of the vertebral column (Fig. 3a–c; Supp. Figure 2a; Supp. Table 2). Notably, we showed that RA constrains TVN in mouse, with an additional 1–2 elements observed following RA depletion for all genotypes where TVN can be quantified, including WT, *Gdf11*[+/−], *miR-196*-TKO and *Gdf11*[+/−];*miR-196*-TKO embryos. This increase manifested as an expansion in the number of presacral elements, the identity of which (see 'Methods') was dependent on the exact allelic combination (Compare Supp. Figure 2a with Fig. 2a). AGN-exposed WT embryos yielded anteriorising transformations of the cervical region (C2 → C1, C7 → C6) at low penetrance, no change in positioning of cervico-thoracic transition (Fig. 3c) and one additional T element (Supp. Figure 2a). All phenotypes were consistent with observed phenotypes in Retinoic acid receptor γ (RARγ) knockout mice[32]. The deletion of even a single *Gdf11* allele under these RA-depletion conditions lead to a higher penetrance of C7 → C6 transformation, and in 40% of embryos, serial transformation resulted in a shift in cervico-thoracic positioning such that 8 cervical elements formed (Fig. 3c). This striking shift in cervico-thoracic positioning, widely considered as one of the strongest evolutionarily constrained traits of mammals[6,33], became almost fully penetrant in *Gdf11*[−/−] embryos

treated with AGN, and where present, usually formed at the expense of a thoracic element and thus was homeotic in nature. Vertebral identity at the cervico-thoracic junction is known to involve the action of *Hox5/6* paralogous groups[34] early in development. Consistent with this, and with observed phenotypic changes, we found the anterior boundary of *Hoxc6* was shifted caudally by 1 somite in E10.5 *Gdf11*[−/−] mutants treated with AGN (Supp. Figure 2b), demonstrating redundancy between RA and Gdf11 in the timely activation/spatial regulation of *Hoxc6*, and likely other trunk *Hox* genes. Countering this displacement of the cervico-thoracic junction, deletion of *miR-196* paralogs returned this major morphological transition back to normal (Fig. 3d, Supp. Table 2), likely through the relief of post-transcriptional suppression of *Hox5/6* targets[16,35] at this site. Nonetheless, additional presacral element(s) still formed in AGN-treated *miR-196*-TKO, *Gdf11*[+/−];*miR-196*-TKO and *Gdf11*[−/−];*miR-196*-TKO embryos, taking on a thoracic or lumbar identity (Compare Fig. 2d with Fig. 3d; Supp. Table 2). Collectively, this extensive genetic/chemical deletion series has revealed that RA, Gdf11 and miR-196 all act to constrain TVN individually, and synergistically. In addition, each factor differentially shapes positional identity along the A-P axis, with individual, synergistic and, in this case, antagonistic interactions revealed. It is important to note that in these embryos, and all embryos assessed in this study, progression through the T-to-T transition always occurred, albeit late. This indicates that the regulatory synergism promoting this key transition was yet to be fully depleted, with prime candidates supporting the eventual T-to-T in compound mutant embryos being Gdf8[36] and potentially FGF signalling[37].

**Factors constraining TVN differentially shape Hox codes.** One shared feature of the extrinsic and intrinsic mechanisms shown here to constrain TVN and shape regional identity is their capacity to influence spatio-temporal *Hox* expression[16,17,38]. Thus, to reveal the full extent to which miR-196 and Gdf11 individually and collectively regulate *Hox* codes during the T-to-T transition (somites 21 +/− 1 [~E9.5]) or after this transition is complete (somite 32 +/− 1 [~E10.5]), we quantified expression of all 39 *Hox* genes within tailbud tissue across the allelic deletion series (Fig. 4). Removal of miR-196 repressive activity (*miR-196a2*[−/−];*b*[−/−], phenotypically equivalent to *miR-196*-TKO) led to a robust upregulation of trunk *Hox* genes and concomitant downregulation, or failure to timely activate, posterior *Hox* genes

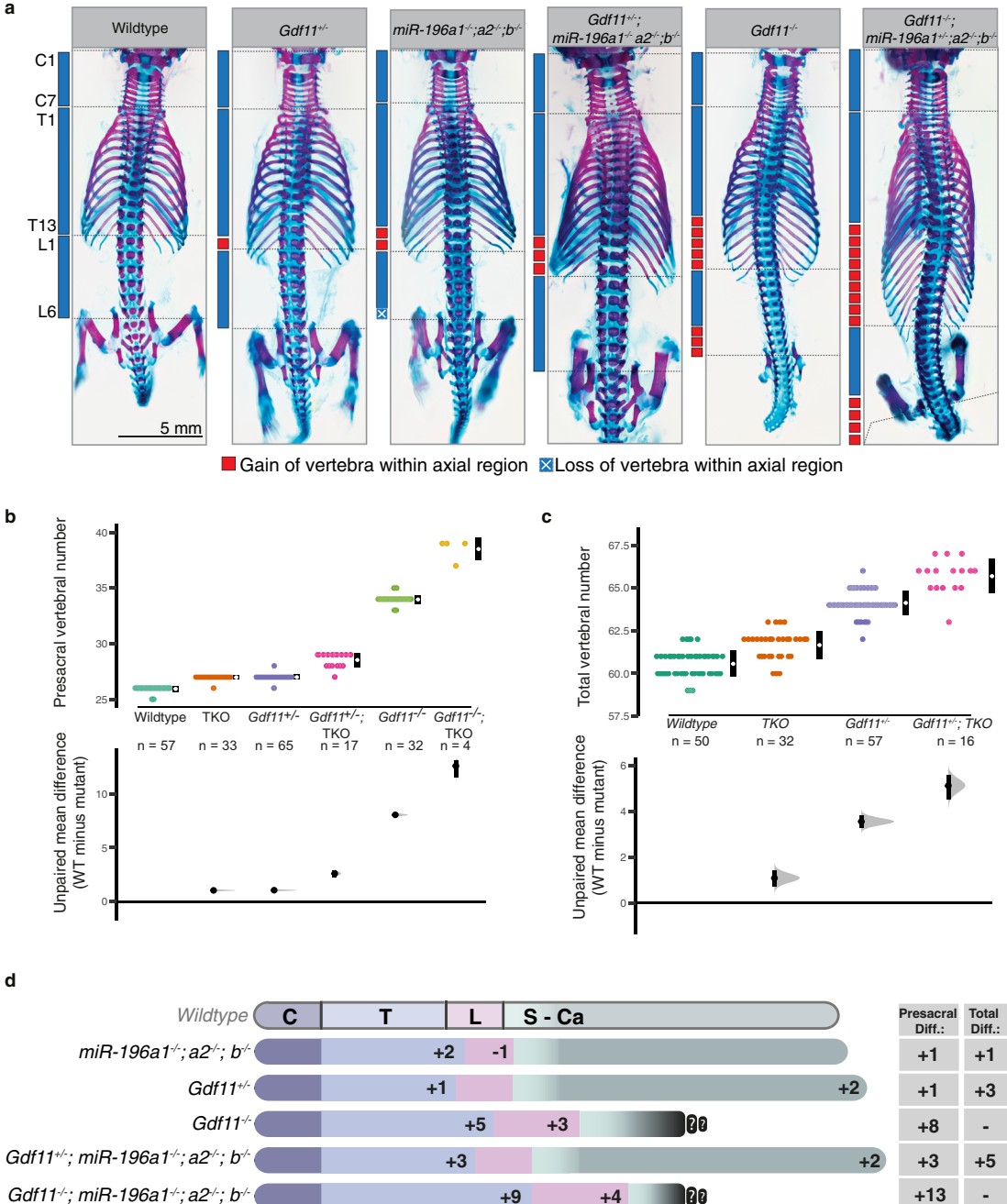

**Fig. 2 miR-196 and Gdf11 act synergistically in vivo to constrain total vertebral number. a–d** In vivo characterisation of *miR-196* and *Gdf11* individual and compound mouse mutant skeletal phenotypes. **a** Representative E18.5 skeletal preparations across genotypes in dorsal view focusing on the presacral vertebral column. C = cervical; T = thoracic; L = lumbar. **b**, **c** Quantification of presacral vertebral number **b**, and total vertebral number **c**, across genotypes. Raw data is presented in the upper plot (vertical error bar = mean and standard deviation). Mean differences relative to shared reference genotype (here WT) are presented in the lower plot as bootstrap sampling distributions. Each mean difference is depicted as a dot and 95% confidence interval is indicated by the ends of the vertical error bar. *n* refers to the number of individual animals used for this analysis. **d** Schematic summary of vertebral alterations observed across the *Gdf11;TKO* allelic deletion series, relative to WT. Numbers represent the unpaired mean difference for a given genotype, rounded to the next whole number. Diff = difference, S-Ca = sacral-caudal vertebrae, question marks indicate dysmorphic and non-quantifiable elements. **b**, **c** Source data are provided as a Source data file.

of all four clusters at E9.5 (Fig. 4)[16], a signature that began to resolve by E10.5 (Fig. 4). From a patterning perspective, either an increased trunk *Hox* code[39] or a delay in activation of Hox10 rib-suppression activity[3,40] have the potential to drive the expanded thoracic identity seen in *miR-196*-TKO embryos. In contrast, loss of one *Gdf11* allele had minimal impact on trunk *Hox* expression, indicating that the mildly expanded thoracic phenotype of

*Gdf11*[+/−] embryos is more likely to result from the modest heterochronic delay in posterior *Hox* code activation seen at E9.5, which largely resolved to WT levels by E10.5 (Fig. 4). This is further supported in *Gdf11*[−/−] embryos, where the level of trunk *Hox* upregulation was not consistent with the dramatically expanded thoracic phenotype, however, a striking reduction in expression of all posterior *Hox* genes was seen at E9.5 (Fig. 4),

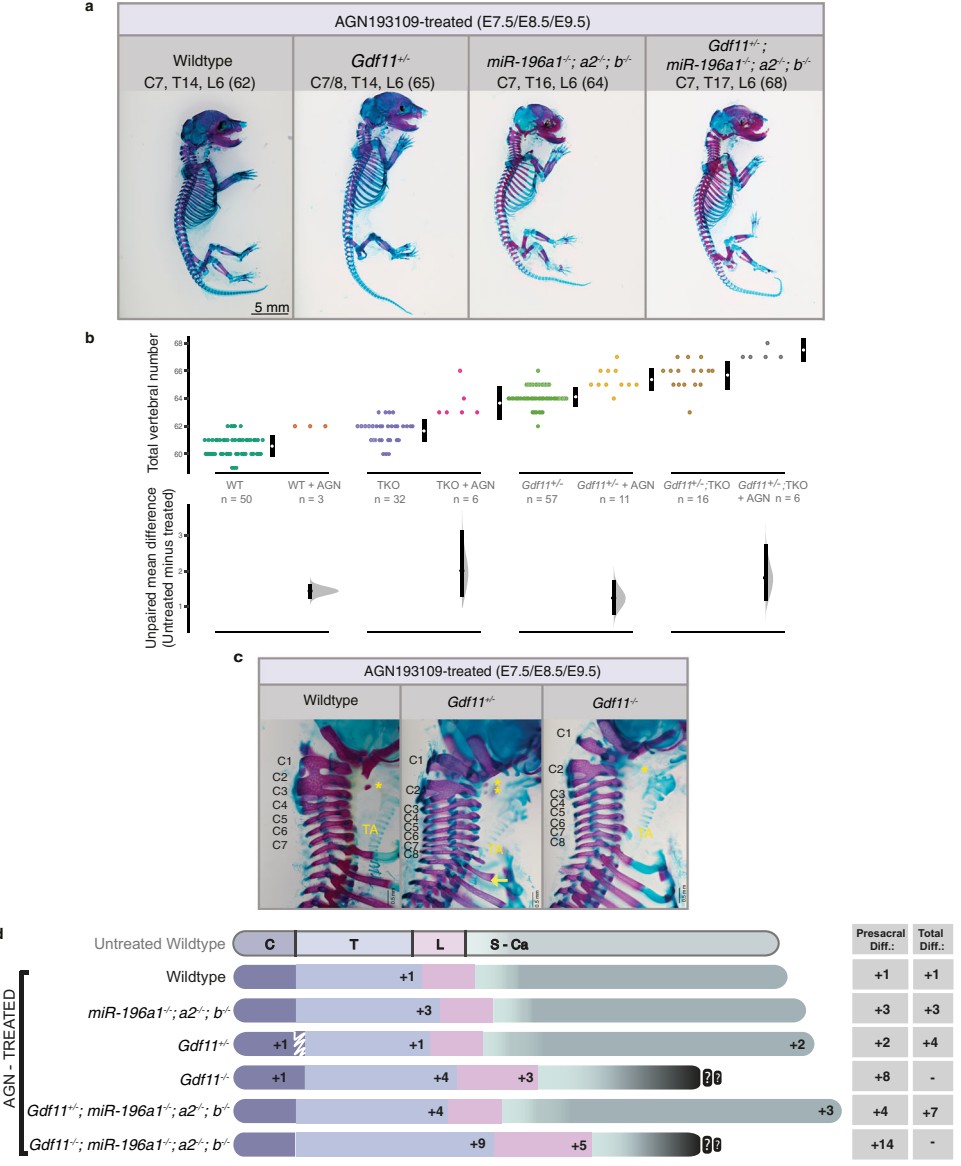

**Fig. 3 Manipulation of multiple regulatory mechanisms is required to further break constraint of mouse axial formulae. a–d** In vivo characterisation of *miR-196* and *Gdf11* individual and compound mouse mutant skeletal phenotypes following treatment with RA-receptor inhibitor AGN193109. **a** Representative embryonic day (E)18.5 skeletal preparations across genotypes. C = cervical; T = thoracic; L = lumbar; total vertebral number indicated in brackets. **b** Quantification of total vertebral number across genotype. Raw data is presented in the upper plot (vertical error bar = mean and standard deviation). Mean differences against the respective untreated control are presented in the lower plot as bootstrap sampling distributions. Mean differences are depicted as dots and 95% confidence intervals are indicated by the ends of the vertical error bars. *n* refers to the number of individual animals used for this analysis. Source data are provided as a Source data file. **c** Altered skeletal identity surrounding the cervico-thoracic boundary in AGN193109-treated WT and *Gdf11* mutant embryos. *=anterior arch of the atlas; TA = tuberculum anterior; yellow arrow indicates a rib anlage on the 8th vertebral element. **d** Schematic summary of vertebral alterations observed across the AGN193109-treated *Gdf11;TKO* allelic deletion series, relative to WT. Numbers represent the maximum phenotypic difference observed from the untreated wild-type control. Diff = difference, S-Ca = sacral-caudal vertebrae, question marks indicate dysmorphic and non-quantifiable elements. White drawn line = partially penetrant, often incomplete or unilateral, alterations observed at the first thoracic element (Supp. Table 2).

with most remaining low at E10.5 (Fig. 4 and Supp Fig. 3a)[15]. Finally, the altered molecular signature revealed following combined loss of all *miR-196* and *Gdf11* alleles reflects the synergistic morphological outcomes observed in presacral regions (Fig. 2a; Supp. Table 1). Collectively, the altered *Hox* codes observed across the allelic deletion series allow us to rationalise the molecular basis for observed patterning changes (Fig. 2d; Supp. Table 1). Moreover, the ability for ectopically-expressed trunk Hox genes *Hoxa5* or *Hoxb8* to drive axial elongation under certain circumstances[30] and the suggested importance of timely

activation of Hox13 paralogs in terminating mouse axial elongation[15,30], raised the question as to whether the heterochronic and quantitative changes in global *Hox* signatures observed here may also underlie changes in TVN.

**Posterior *Hox* expression supports tail formation.** The understanding of Hox12 and Hox13 paralog function during vertebral column formation is incomplete at best due to the current lack of paralog group mouse knockouts, and recently, the view of Hox13

**Fig. 4 Quantitative changes in *Hox* cluster expression downstream of individual and cumulative Gdf11 and miR-196 deletion.** Differential gene expression analysis of all 39 *Hox* genes in somite-matched tailbud samples of the *Gdf11;TKO* allelic deletion series compared to WT. For each mutant genotype, individual *Hox* genes are normalised against the WT condition. Grey colour = no expression or below detection threshold, Ct = threshold cycle. Source data are provided as a Source data file.

genes as endogenous axis terminators has been challenged in fish[41]. To interrogate posterior Hox function, and specifically, to dissect whether the delay in/loss of posterior *Hox* activation seen in *Gdf11*$^{+/-}$ and *Gdf11*$^{-/-}$ mutant embryos is of any phenotypic consequence, we sought to restore timely posterior *Hox* expression through the creation of transgenic mice expressing either *Hoxd11* (*Hoxd11*$^{OE}$) or *Hoxd12* (*Hoxd12*$^{OE}$) under the control of *Cdx2* regulatory elements[42] (Supp. Figure 4a-b).

Initial characterisation of these lines on an otherwise WT background revealed a reduction in lumbar count by 1 element and concomitant reduction in TVN, with both qualitative (*Hoxd12* > *Hoxd11*) and quantitative (copy number) enhancement of phenotype observed (Fig. 5a–c; Supp. Figure 4c-d; Supp Table 3). This very mild reduction phenotype of the presacral region was surprising when compared with full tail truncation observed following ectopic expression of Hox13 paralogs[15,30], which may stem from quantitative differences between transgenics or likely functional differences between these Hox proteins[43]. A near-identical reduction in lumbar count/TVN was observed when either *Hoxd11*$^{OE}$ or *Hoxd12*$^{OE}$ was bred onto a *Gdf11*$^{+/-}$ background (Fig. 5a–c; Supp. Figure 4c,e; Supp. Table 3), supporting the view that the temporary delay in posterior Hox activation in these heterozygous embryos (Fig. 4) may indeed contribute to a shifted T-to-T transition and elongation phenotype, or at a minimum, that ectopic posterior *Hox* expression is dominant over elongation mechanism(s).

Each of the above genetic crosses represents a cumulative *Hox* scenario that is ectopic over WT levels, and thus we next cross-bred either *Hoxd11*$^{OE}$ or *Hoxd12*$^{OE}$ onto the *Gdf11*$^{-/-}$ background, which is greatly depleted for posterior *Hox* expression (Fig. 4). Relative to truncated *Gdf11*$^{-/-}$ embryos, restoration of a single posterior *Hox* gene yielded striking restoration of a segmented *Uncx4.1*$^+$ tail-like structure at E12.5 and E13.5, albeit ventrally displaced (Fig. 6a; Supp. Figure 5a-b). Detailed characterisation of *Gdf11*$^{-/-}$ embryos via tissue sections (Supp. Figure 6a) revealed in many cases that the notochord underlying the primary neural tube had turned abnormally relative to WT, projecting ventrally at the level of the cloaca. There, notochordal cells were seen interspersed with Sox2$^+$-neural and enveloped by Foxa2$^+$-endodermal cells, with no observable patterning or structure to this previously noted ventral mass[44,45] (Supp. Figure 6a [v–vii]). The restoration of either *Hoxd11* or *Hoxd12* expression in *Gdf11*$^{-/-}$ embryos did not prevent ventral projection of the notochord, however, these cells no longer became trapped and extended to the tip of the ventral tail in an organised manner (Fig. 6b, Supp. Figure 6a [ix–xii], [Supplementary movies 1-3]). Adjacent to the notochord in these *Gdf11*$^{-/-}$;*Hoxd11/12*$^{OE}$ embryos, an organised Foxa2$^+$-tailgut formed, extending from the cloacal orifice to the tip of the ventral tail, with distal cells co-expressing Foxa2 and Sox2 (Fig. 6b, Supp. Figure 6b, [Supplementary

movies 1-3])[46]. One signature component of a wild-type tail that could not be clearly delineated in this ventral tail was a Sox2$^+$ secondary neural tube, despite the tailbud housing all major progenitors required for posterior body formation based on marker analysis (Fig. 6c, Supp. Figure 6b). Collectively, the presence of a patterned tail structure in *Gdf11*$^{-/-}$;*Hoxd11/12*$^{OE}$ embryos, and the maintenance of axial progenitors in this ventrally-displaced tailbud (Fig. 6c) at stages when these progenitors should normally be close to exhaustion[47], strongly supports a minimum requirement for posterior *Hox* expression as an integral component of the gene regulatory network necessary to construct the tail. Despite this striking result at mid-gestation, skeletal analysis of E18.5 *Gdf11*$^{-/-}$;*Hoxd11*$^{OE}$ or *Gdf11*$^{-/-}$;*Hoxd12*$^{OE}$ embryos revealed no vertebrae arising from ventrally-displaced *Uncx4.1*$^+$-somites (Supp. Figure 5b). In fact in many *Gdf11*$^{-/-}$;*Hoxd12*$^{OE}$ embryos, the last-formed vertebra was found more rostrally than in *Gdf11*$^{-/-}$ embryos alone (Supp. Figure 5c), likely due to the prolonged expression of posterior *Hox* genes resulting in increased apoptosis, decreased proliferation and degeneration of more mature tail structures[48] and/or a lack of secondary neurulation. In this light, these mutant embryos were more akin to human development, where a patterned tail is initially generated but fully regresses over time[49].

## Discussion

Historical hereditary studies have long supported a multigenic contribution to even mild examples of intraspecies vertebral variation[50]. Here, we reveal the highly integrated manner by which multiple regulatory layers constrain the modular logic of the vertebral column. Importantly, we were able to produce viable offspring with 5 additional vertebral elements spanning both trunk and tail regions, and with a maximal expansion of 7 additional elements. Regarding the latter, we expect that spatial and/or temporal conditional deletion of these regulatory mechanisms would circumvent embryonic lethality, enabling a vastly altered mammalian body plan, though the potential secondary consequences of this manipulation on locomotion or internal organ formation cannot be predicted. Combined with the recent identification of the Lin28/let-7 axis in constraining vertebral number in the tail[14,15], these results demonstrate the depth of genetic redundancy that normally acts to constrain each vertebral region in the mouse and provide robustness to the system as a whole.

The comprehensive allelic deletion series performed allowed us to investigate how the breaking of regional constraint within this experimental model aligns with current hypotheses of natural variation. For example, the break in C7 constraint observed in sloths has been suggested to result from a mis-alignment of somite-derived (primaxial) vertebral elements with that of lateral plate mesoderm-derived (abaxial) distal rib/sternum, such that 7

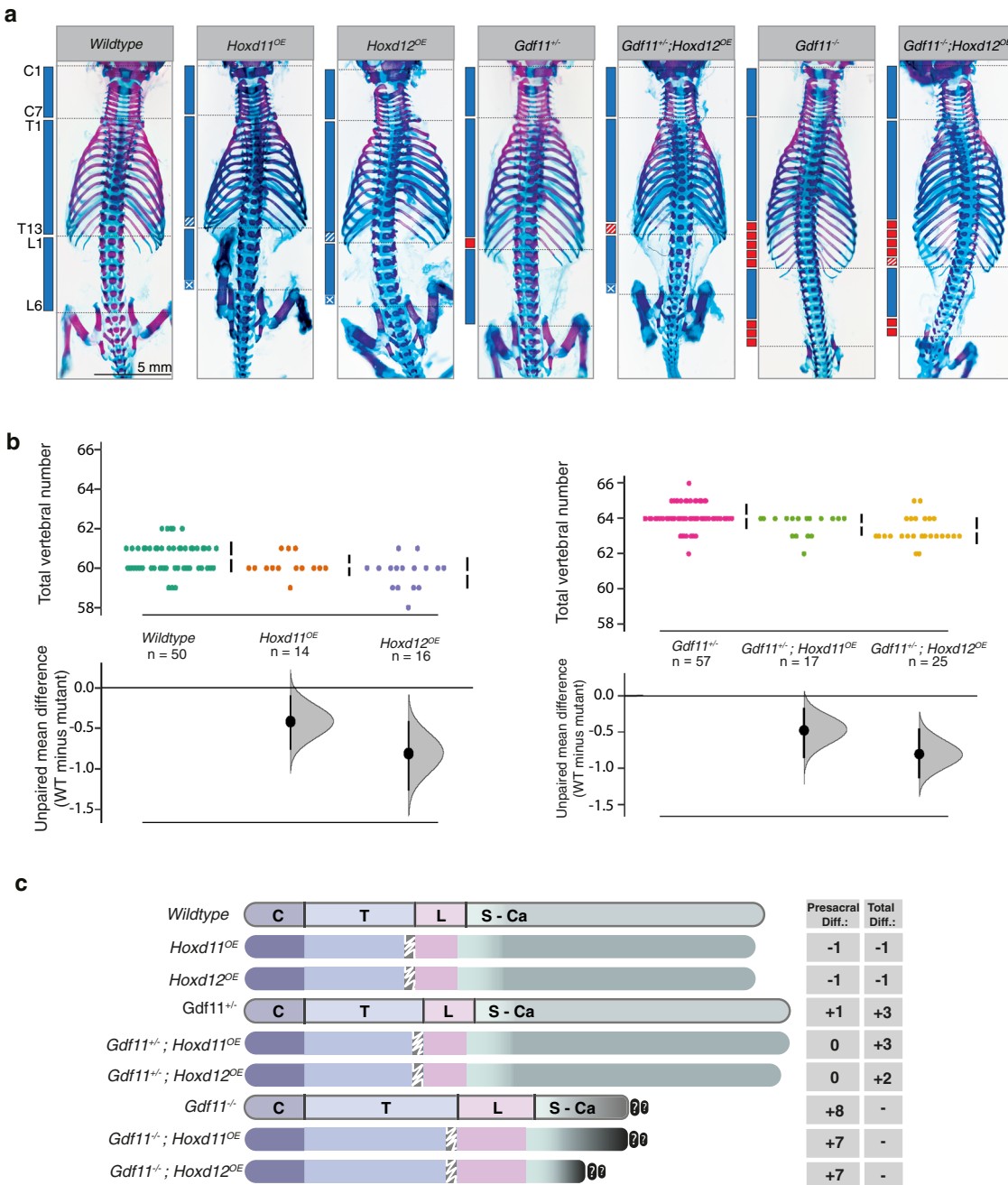

**Fig. 5 Qualitative and quantitative Hox functions in the control of lumbar count and total vertebral number. a–c** Transgenic expression of *Hoxd11* or *Hoxd12* using the *Cdx2* promoter is able to partially reverse anterior transformations of the lumbar region in *Gdf11*+/− and *Gdf11*−/− mutant mice, and reduce total vertebral number in WT and *Gdf11*+/− animals. **a** E18.5 skeletal analysis of *Hoxd11*OE or *Hoxd12*OE (OE = overexpressor) intercrossed with the *Gdf11* mutant line revealed changes in axial formulae. C = cervical; T = thoracic; L = lumbar. **b** Quantification of total vertebral number (TVN) in WT and *Gdf11*+/− embryos, with or without the presence of *Hoxd11*OE and *Hoxd12*OE transgenes. Raw data is presented in the upper plot (vertical error bar = mean and standard deviation). Mean differences relative to WT are presented in the lower plot as bootstrap sampling distributions. Each mean difference is depicted as a dot and 95% confidence interval is indicated by the ends of the vertical error bar. *n* refers to the number of individual animals used for this analysis. Source data for **b** are provided as a Source data file. **c** Schematic summary of changes in axial formulae, relative to WT, observed in single *Hoxd11*OE or *Hoxd12*OE transgenic lines, and following cross-breeding of each transgenic with the *Gdf11* mutant line. Numbers represent the rounded (to the next whole number) unpaired mean difference for a given genotype. C = cervical, T = thoracic, L = lumbar, S-Ca = sacral-caudal, question marks indicate dysmorphic and non-quantifiable elements. White drawn line = frequently observed reduction/malformation of ribs on the last rib-bearing thoracic element in mice carrying the *Hoxd11*OE or *Hoxd12*OE transgene.

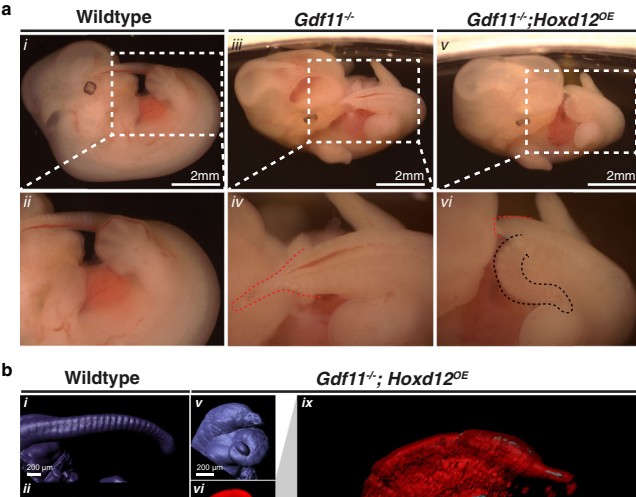

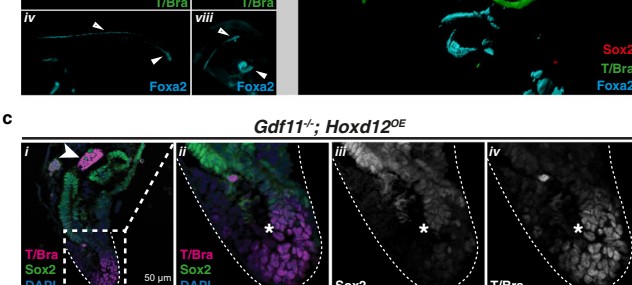

**Fig. 6 Posterior Hox genes can support axial elongation. a** Morphological analysis of tail phenotypes in E12.5 WT (i–ii), *Gdf11*$^{-/-}$ (iii–iv) and *Gdf11*$^{-/-}$; *Hoxd12*$^{OE}$ (v–vi) mouse embryos. Brightfield images of whole embryos, with tail region enlarged for clarity. Red dotted line demarcates the caudal truncation phenotype observed in the majority of *Gdf11*$^{-/-}$ embryos, ending in a thinning blood-filled spike (iv) (*n* = 41/52). Black dotted line demarcates the ventral tail observed in the majority of *Gdf11*$^{-/-}$;*Hoxd12*$^{OE}$ embryos (vi) (*n* = 22/32). **b** 3D reconstruction of caudal structures within E12.5 WT (i–iv) and *Gdf11*$^{-/-}$; *Hoxd12*$^{OE}$ (v–ix) embryos immunostained for Sox2 (red), T/Brachyury (T/Bra; green) and Foxa2 (cyan). Enlarged and merged image of channels presented in (vi–viii) is shown in (ix). Tissue-restricted expression of Foxa2 within the ventral neural tube or tailgut is indicated using a white arrowhead. **c** Section immunofluorescence analysis reveals axial progenitor populations are present in the tailbud of *Gdf11*$^{-/-}$;*Hoxd12*$^{OE}$ ventral tail at E12.5 (*n* = 2). Co-staining for T/Bra (magenta) and Sox2 (green) identified single-positive cells of either marker, along with dual positive cells indicative of neuromesodermal progenitors (asterisk). All sections are stained for DAPI (blue), arrowhead indicates notochord. *n* refers to the number of individual animals used for this analysis.

cervical vertebral bodies always form and axial diversity arises due to the gain or loss of ribs[8,9]. Here, in RA-depleted *Gdf11*$^{-/-}$ embryos with 8 cervical elements, we observe serial anteriorising homeotic transformations for at least 2 elements on either side of the C-T boundary supporting a genuine increase in cervical number. However, in the majority of RA-depleted *Gdf11*$^{+/-}$ embryos where 7 cervical elements formed, we see that the pattern of centra ossification of the first rib-bearing element is more indicative of a cervical element, and T2 is displaced posteriorly. This suggests that loss of even one Gdf11 allele in this context has

the ability to preferentially repattern primaxial tissue, and supports the mis-alignment of primaxial and abaxial tissues as a mechanism to maintain constraint (here) or drive diversity (in sloths).

At a molecular level, the 3′-to-5′ sequential activation of *Hox* cluster expression within axial progenitors over development time —the *Hox clock*[51]—prefigures Hox spatial collinearity along the A-P axis[52,53], though the consequences of clock manipulation are not easy to address. The mouse mutants presented here, all unified in their temporal control of global *Hox* transitions, offer a granular view as to the consequences of manipulating that *Hox* clock in vivo. We show that the speeding up, or the slowing down, of *Hox* cluster signatures result for the most part in serial homeotic transformations that associate with changes in total vertebral number. At a minimum, this implies a very tight association between *Hox* patterning and elongation mechanisms, though importantly, our data also revealed that posterior *Hox* genes have the capacity to positively influence axial elongation. This latter data, viewed along with similar results for trunk Hox genes in mouse[30] and *Hox13* paralogs in zebrafish[41], allows us to propose a model whereby multiple post-occipital expressing Hox paralogs participate in construction of the main body axis, not solely in its patterning. We do not suggest that *Hox* genes are the primary drivers of axial elongation, but that a minimum level is required. This model does not preclude a role for high levels of posterior *Hox* expression in terminating axial elongation[15,30,43], and indeed our data on ectopic *Hoxd11*/*Hoxd12* can be viewed as supporting this. However, caution should be taken when interpreting overexpression studies[41] while awaiting the genetic deletion of *Hox12* and *Hox13* paralog groups in the mouse. Why then has a role for *Hox* genes in shaping vertebral number not been apparent in the extensive *Hox* mouse mutant literature? Cumulative mutant analysis has shown that each individual vertebral element is patterned by at least two, if not more, Hox paralog groups[34], thus only by removal of multiple adjacent paralog groups would this function be revealed. Experimentally, this would be a complex undertaking and of questionable relevance from an evolutionary perspective. However, by altering the timing of collective *Hox* code transitions (i.e. in vivo pacing of the *Hox* clock) rather than Hox function per se, through changes in widespread post-transcriptional regulation (miR-196) or alterations in higher-order signalling (Gdf11 haploinsufficiency), this unanticipated vertebrate *Hox* function has now been revealed.

From a cellular perspective, we show here the persistence of axial progenitors in the tailbud of *Gdf11*$^{-/-}$;*Hoxd12*$^{OE}$ embryos beyond what is seen in WT. Whether the restoration of posterior Hox expression in this scenario is acting cell-autonomously to maintain progenitor proliferation and support tail formation remains to be elucidated, though recent work in zebrafish argues against such a role[41]. Serial transplantation of axial progenitors in vivo[54] has shown that these cells do not intrinsically exhaust following a timer mechanism, highlighting the importance of extrinsic signals in the maintenance and exhaustion of a progenitor pool. In this light, the restoration of tailgut formation along the full extent of the *Gdf11*$^{-/-}$;*Hoxd12*$^{OE}$ ventral tail was of particular interest, since surgical removal of the caudal endoderm in chick has been shown to redirect the notochord ventrally into the hindgut[55] with striking similarity to what is seen in *Gdf11*$^{-/-}$ mutants. Moreover, expression of a dominant-negative Hoxa13 protein within chick caudal endoderm, but not mesoderm, resulted in tail truncation[55], further supporting the positive requirement for posterior Hox function in tail construction across multiple species, and highlighting important cellular targets for future investigation.

By focusing on how one mammalian species cannalises axial formulae, this work has reshaped the current understanding of

body plan formation and the critical function of Hox networks within. This should help inform the basis for supernumerary vertebral elements that have been reported to appear in ~3% of the human population[56], constituting congenital anomalies such as an 8th cervical vertebra[57], additions of up to 3 thoraco-lumbar vertebrae[58–60] and rare cases of humans with tail vertebrae[61]. Moreover, the comprehensive allelic deletion series performed here provides us with a window into how evolutionary changes may have occurred, not by complete ablation of one signal or regulatory mechanism, but by more subtle and/or tissue-restricted changes in multiple pathways that converge in their ability to shape axial formulae.

## Methods

**Animal experimentation ethical approvals**. All animal procedures were performed in accordance with the Australian Code of Practice for the Care and Use of Animals for Scientific Purposes (2013). Experiments were approved by the Monash Animal Ethics Committee under project numbers MARP/2011/012, MARP/2015/168 and MARP/2015/123.

**Mice**. $miR\text{-}196a1^{GFP}$, $miR\text{-}196a2^{GFP}$, $miR\text{-}196a1^{-/-}$, $miR\text{-}196a2^{-/-}$ and $miR\text{-}196b^{-/-}$ mouse lines have been previously described[16], and were maintained on a C57BL/6 (mixed substrain J/N) background. $Gdf11^{-/-}$ mutant mice have been previously described[17] and were maintained on a C57BL/6J background. $Cdx2P{:}Hoxd11$ ($Hoxd11^{OE}$) and $Cdx2P{:}Hoxd12$ ($Hoxd12^{OE}$) transgenic mice were generated in-house by pronuclear injection into C57BL/6J zygotes according to standard protocols[62]. cDNA for mouse $Hoxd11$ or $Hoxd12$ was inserted downstream of the 9.4 kb $Cdx2$ regulatory region ($Cdx2P$)[42]. Founder animals were verified by standard PCR, copy number assessed by digit drop PCR and germline transmission confirmed ($Cdx2P{:}Hoxd11$ 1 line established; $Cdx2P{:}Hoxd12$ 2 lines established).

**Oral gavage**. The pan retinoic acid receptor inhibitor AGN193109 (Tocris) was administered by oral gavage to pregnant dams at a concentration of 0.8 mg/kg on 3 consecutive days of development (E7.5, E8.5 and E9.5). Stock AGN193109 was resuspended in DMSO (2 mg/mL), stored at −20 °C and freshly diluted in corn oil (1:25) on the first day of administration.

**Skeletal preparation and imaging**. Skeletal preparation was performed on E18.5 embryos or p0 postnatal pups as previously described[63]. Embryos or pups were skinned and eviscerated, followed by sequential 2-day incubations in 100% ethanol, 100% acetone and stain solution (containing 0.15 mg/ml Alican blue, 0.5 mg/ml alizarin red, 70% ethanol and 5% glacial acetic acid), with constant rocking at room temperature (RT). Soft tissue matter was cleared from the skeleton by incubation in 1% potassium hydroxide (KOH) for 2–5 days. Once sufficiently cleared, samples were subjected to an increasing glycerol/1% KOH gradient (25%, 50%, 75%, 100% glycerol) for a minimum of one day each. Samples were stored in 100% glycerol. Skeletal images were taken with the Vision Dynamic BK Lab System at Monash University in the Paleontology Lab on a Canon 5d MkII with a 100 mm Macro lens (focus stop 1:3/1:1). To extend the focal depth on the skeletal images multiple images were taken and stacked in ZereneStacker using the PMax algorithm.

**Skeletal phenotyping and statistical analysis**. Minor differences in axial formulae have been reported between mouse strains, and thus it was of critical importance that all analyses were performed on an isogenic C57BL/6 background, decreasing inter-animal variation and allowing robust quantification. For each skeletal preparation, axial formulae were assessed independently by two individuals, blinded to genotype. Where unilateral identity changes were observed, the vertebral identity that deviates from WT was annotated, e.g. unilateral L1 rib would be annotated as a L1-to-T transformation. In skeletons that have been exposed to AGN193109, we commonly observed a dorsal bifurcation of C2 which fused laterally into one and therefore were counted as one vertebral element. Unpaired mean differences in total vertebral number counts are presented in Supplementary Table 4. Animal sex has not previously been shown to alter skeletal formulae and thus was not assessed. For statistical analysis and data visualisation of vertebral numbers, the R-package "Data Analysis using Bootstrap-Coupled ESTimation" (DABEST)[64] was used. To determine mean differences to the respective shared wild-type control 5000 bootstrap samples were taken and the confidence interval was bias-corrected and accelerated.

**Droplet digital PCR assay and copy number detection**. Each droplet digital (dd) PCR assay contained 20 ng of DNA template, 1×ddPCR Master Mix (Bio-Rad, USA), 0.5 µM primer-F, 0.5 µM primer-R and 0.25 µM probes. Primer and probe sequences are shown in Supplementary Information. A QX200 ddPCR droplet generator (Bio-Rad, USA) was used to divide each 20 µL ddPCR mixture into

approximately 20,000 droplets. The ddPCR probe for the reference sequence ($Rpp30$) was labelled with hexachlorofluorescein (HEX) at the 5′ end and black hole quencher (ZEN/IowaBlack) at the 3′ end. The ddPCR probe for the target sequence ($Hoxd12$) was labelled with 6-carboxyfluorescein (FAM) at the 5′ end and black hole quencher (ZEN/IowaBlack) at the 3′ end. The thermal parameters were as follows: 10 min at 95 °C, followed by 40 cycles of 30 s at 94 °C and 1 min at 60 °C, followed by 98 °C for 10 min. The amplified products were analysed using a QX200 droplet reader (Bio-Rad, USA) and copy number analysed using QuantaSoft Analysis Pro Version 1.0 (Bio-Rad, USA).

**In situ hybridisation**. Whole-mount in situ hybridisation was performed as previously described[65] with some modifications. Embryos were dissected in ice-cold PBS (Gibco, 14190-144) and tissue overlying the hindbrain pierced. Embryos were fixed in 4% paraformaldehyde (PFA), rocking overnight at 4 °C. All steps were performed with gentle rocking at RT unless otherwise specified. Embryos were washed twice with PBS-Tween (0.1%, PBT) for 5 min. Embryos were dehydrated using a methanol/PBT gradient (25%, 50%, 75%, 100% methanol) with each step ranging from 5-20 min depending on age. Embryos could be stored in 100% methanol, or to begin in situ hybridisation, embryos were rehydrated following a reversed methanol/PBT series as above, washed twice with PBT for 5 min and treated with 10 µg/ml of proteinase K in PBT (E9.5 for 8 min, E10.5 for 15 min and E12.5 for 25 min). Embryos were washed in PBT twice for 5 min and post-fixed in 4% PFA with 0.2% glutaraldehyde for 20 min. Following this, embryos were washed in PBT twice for 5 min and put into hybridisation solution (50% formamide; 5x SSC (pH4.5), 1% SDS; 50 µg/ml heparin; 50 µg/ml yeast tRNA (Sigma, R6750)) at 70 °C for a minimum of 2 h before addition of 1 µg/ml DIG-labelled riboprobe and incubated overnight rocking at 70 °C. The next day, embryos were washed in Solution I (50% formamide; 5x SSC (pH4.5); 1% SDS) three times for 30 min each, with gentle rocking at 70 °C. Embryos were then washed in Solution II (50% formamide; 2x SSC (pH4.5); 0.1% Tween-20) three times for 30 min each, gently rocking at 65 °C. Embryos were washed in TBS-T (0.1% Tween) three times for 5 min at RT and transferred to blocking solution (TBS-T containing 10% heat-inactivated sheep serum) for 2 h at RT. Embryos were then placed into blocking solution containing 1:2000 anti-DIG antibody (Roche, 11093274910) rocking overnight at 4 °C. The next morning, embryos were washed at least five times in TBS-T for 1 hr each time, and again overnight at 4 °C. The following day, embryos were washed in NTT (100 mM NaCl, 100 mM Tris-HCl (pH9.5); 0.1% Tween-20) 3 times for 10 min. For colour development, embryos were incubated in BM purple (Roche, 11442074001), protected from light. Colour development was stopped by washing 3 times in PBT for 5 min each, and post-fixed in 4% PFA for 20 min. Embryos were rinsed in PBT 3 times for 5 min each and stored in PBT at 4 °C until imaged. Plasmids for riboprobe generation were kind gifts from A. Mansouri ($Uncx4.1$), P. Sharpe ($Hoxc6$), C. Tabin ($Hoxa11$, $Hoxd11$, $Hoxd2$) or were generated in-house ($Hoxc12$, $Hoxb13$).

**Cell lines**. The Bruce4 mouse embryonic stem cell line was originally isolated from a C57BL/6 mouse[66] and kindly provided by Dr Jeff Mann. Two in-house generated iPSC lines were derived from the miR-196 triple knockout mouse strain[16] and wild-type mice of an isogenic background.

**Generation of mouse induced pluripotent stem cells (iPSCs)**. iPSCs were generated from tail tip fibroblasts of adult WT and $miR196a1^{gfp/gfp}, a2^{gfp/gfp}, b^{-/-}$ mice as previously described[24]. Briefly, tail tip fibroblasts were isolated and plated with standard fibroblast media containing DMEM, 10% FBS, 1x Glutamax (Gibco 35050061) and 1x Pen/Strep (Gibco 15140122). After confluency, fibroblasts were infected with a Doxycycline-inducible lentivirus expressing the OKSM cassette. Reprogramming was carried out for 16 days following the addition of Doxycycline. Clonal lines were established after withdrawal from Doxycycline and maintained in ES media containing DMEM KO, 15% ES grade FBS, 1x Glutamax (Gibco 35050061), 1x Pen/Strep (Gibco 35050061), 1x NEAA and 100 mM β-mercaptoethanol and 1000 U/ml LIF on mitotically inactive mouse embryonic fibroblasts. Established iPSCs lines were passaged every 72 h at low density.

**Teratoma assay**. Pluripotency of individual iPSC lines were tested by assessing their ability to form teratomas in vivo[24]. Mouse iPSCs were trypsinized and feeder-cell depleted twice for 20 min. Approximately $1 \times 10^6$ WT or $miR\text{-}196\text{-TKO}$ iPSCs were subcutaneously injected into the dorsal flank of isoflurane-anaesthetised immune-suppressed NOD-SCID mice of 6–8 weeks of age. Within 4–6 weeks, teratoma formation was evident and consequently removed, washed in Dulbecco's PBS and fixed in 4% paraformaldehyde (Merck Millipore, 30525-89-4) overnight at room temperature (RT). Teratomas were then embedded in paraffin, sectioned at 5 µm and stained with hematoxylin and eosin. Slides were scanned using MIRAX Scan software and viewed with MIRAX Viewer 1.12 software.

**Differentiation of mouse iPSCs and ESCs**. Mouse iPSCs and ESCs were differentiated according to published protocols[25,26] with minor modifications. Briefly, iPSCs or ESCs were routinely maintained in ES media (Composition: 81.8% Knockout DMEM (Gibco, 10829-018); 15% foetal bovine serum; 1% Pen Strep (Gibco, 15140-122); 1% GlutaMAX-I (Gibco, 35050-061); 1% MEM NEAA (Gibco,

11140-040), 0.2% 1000×2-Mercaptoethanol (Gibco, 21985-023)) supplied with LIF (1000x, made in-house) on mitotically inactive primary mouse embryo fibroblasts. In preparation for differentiation, cells were detached in 0.25% Trypsin-EDTA (Gibco, 25200-056), feeder depleted once for 45 min and plated on gelatin-coated (0.1% Gelatin Sigma, G1890-100G) 6-well plates (Falcon, 353046) at a density of $8 \times 10^3$ cells/cm$^2$ in ES media. Cells were given 24–48 h to settle. To start differentiation (Day (D) 0), ES media was changed to N2B27 media (1:1 medium of Advanced Dulbecco's Modified Medium F-12 (Gibco, 12634028) supplemented with 1x N2 (Gibco, 17502001), and Neurobasal medium (Gibco, 21103049) supplemented with 1x B27 with or without Vitamin A, 1x Glutamax (Gibco, 17504044 or 12587010), 40 µg/ml BSA Fraction V (Gibco, 15260037) and 100 mM $\beta$-mercaptoethanol. For ESC differentiation, N2B27 was supplemented as follows: D0—10 ng/ml bFgf (human, Miltenyi Biotec), refreshed at 24 h; D2—5 µM CHIR99021 (StemMACS, 130-103-926); D6—50 ng/ml Gdf11 (human, Miltenyi Biotec). Cells were mechanically dissociated and split at a ratio 1:3 on D3 of differentiation. For iPSC differentiation, N2B27 was supplemented as follows: D0—20 ng/ml bFgf (human, Peprotech); D2—10 µM CHIR99021 (StemMACS, 130-103-926); D5—50 ng/ml Gdf11 (Preprotech), with cells reseeded on D6.

**Tissue microdissection**. The entire presomitic mesoderm and adjacent tissue, caudal to the most recent formed somite, was collected from E9.5 and E10.5 embryos. Presomitic mesoderm tissue was immediately placed in RTL lysis buffer (Qiagen), frozen on dry ice and stored at −80 °C. Yolk sac tissue was collected and used for genotyping where required. The remainder of each embryo was fixed overnight in 4% paraformaldehyde at 4 °C and processed for whole-mount in situ hybridisation to detect *Uncx4.1* expression and determine somite number.

**RNA extraction**. RNA, from cells undergoing in vitro differentiation or cells isolated from in vivo tissue microdissection, was extracted using RNAeasy micro (QIAGEN) or Nucleospin RNA (Macherey-Nagel GmbH & Co) kits.

**Gene expression analysis by BioMark Fluidigm**. 100 ng (E9.5 and E10.5) tailbud RNA was used for cDNA synthesis performed with RT-Vilo (ThermoFisher). Quantitative PCR was performed using the 96×96 BioMark Fluidigm format. Taqman probes used are listed in Supplementary Table 5. Raw Ct values were analysed using a modified version of the qPCR-Biomark script [https://github.com/jpouch/qPCR-Biomark] and normalised as described[67]. In brief, only Ct-values in the optimal range for the Biomark system of 6–25 were used for further analysis. All genes were first normalised against the mean raw Ct-values of five housekeeping gene probes yielding ΔCt values, then normalised against the wild-type condition yielding ΔΔCt values. Finally, the log2(ΔΔCt)-values for genes of interest were visualised as heatmaps using ggplot2 (v3.3.2)[68].

**Gene expression analysis by quantitative PCR**. Additional quantitative PCR analysis was performed on a Lightcycler 480 (Roche) using the SYBR Green I Master Mix (Roche). Two microlitres of cDNA was amplified per reaction using the following programme: 95 °C for 10 s (1 cycle), 95 °C for 10 s, 60 °C for 15 s, 72 °C for 10 s (45 cycles). Primers used in these studies were either previously published[69] or synthesised in-house and listed in Supplementary Table 6.

**Dissection and tissue processing for sectioning**. Whole mouse embryos from various stages were dissected and fixed overnight in 4% PFA/PBS at 4 °C. Samples were washed twice for 10 min in PBS at 4 °C. Samples were equilibrated through a series of sucrose solutions (5%, 20%, 30% sucrose in PBS) before being embedded in OCT medium (TissueTek) using an ethanol-bath and stored at −80 °C. Frozen tissue was sectioned at 12–14-µm thickness on a Leica cryotome. Sections were mounted directly onto slides (Superfrost, Fisher Scientific).

**Section immunofluorescence**. Slide-mounted sections were performed as previously described[70]. Slides were placed into blocking solution (5% heat-inactivated goat serum, 0.5% bovine serum albumin, 0.1% Triton-X in PBS) for at least 30 min. Primary antibodies were added into blocking solution at the concentrations specified below, placed onto the slides, coverslipped and incubated overnight at 4 °C in a humidified chamber. The following day, coverslips were removed and slides were washed in PBS + 0.1% Triton-X (PBSTx) 3–5 times for at least 5 min each. Secondary antibodies were added to the blocking solution at 1:1000, placed onto the slides, coverslipped and incubated for 1 h at RT in a humidified chamber. Slides were washed in PBSTx 3–5 times for at least 5 min. One wash included DAPI (1:1000). Slides were mounted in Prolong Gold Mounting Media. Primary antibodies: rabbit anti-T/Brachyury (T) antibody (AbCam, Ab209665) and rat anti-SOX2 antibody (AbCam, Ab92494), rabbit anti-Foxa2 (SevenHills Bioreagents, WRAB-1200) were used at 1:1000. Secondary antibodies: anti-rat AlexaFluor 488 (ThermoFisher, A212208) and anti-rabbit AlexaFluor 555 (ThermoFisher, A31572) were used at 1:1000.

**Whole-mount immunofluorescence**. E12.5 embryos were dissected in PBS, fixed overnight in 4% PFA at 4 °C on a rocking table. Embryos were then washed in PBS + 0.1% Tween20, dehydrated through a methanol/PBS-Tween series, and stored at −20 °C until further processing. Whole-mount immunofluorescence was performed following a published protocol[71], with minor modifications. Briefly, embryos were bleached in methanol + 6% $H_2O_2$ overnight at 4 °C, rehydrated through a methanol/PBS-Tween series and blocked for at least 24 h in PBS + 0.2% Gelatin + 0.5% Triton X100 (PBS-GT). Embryos were incubated with the following primary antibodies in PBS-GT for a minimum of 5 days: goat anti-T/Brachyury (1:100, R&D, AF2085), rat anti-Sox2 (1:250, Fisher Scientific, Btjce) and rabbit anti-Foxa2 (1:500, SevenHills Bioreagents, WRAB-1200). After antibody exposure, all washes were performed with PBS + 0.5% Triton X100 (PBS-T) at least 6 times over the course of 24 h. Primary antibodies were detected sequentially: first with a donkey anti-goat-AlexaFluor-647 (1:1000, ThermoFisher, A-21447) for 3 days in PBS-GT, washed again as above, then with a goat anti-rat-AlexaFluor-555 (1:1000, ThermoFisher, A-21434) and a goat anti-rabbit-AlexaFluor-790 (1:1000, Invitrogen, A11369) for another 3 days in PBS-GT. After another series of washes, embryos were cleared in 50% tetra-hydrofurane (THF)/50% $H_2O$ overnight, 80% THF/20% $H_2O$ for 2 h, 100% THF for 2 h, twice, 100% di-chloromethane (DCM) for 1 h, and 100% di-benzyl-ether (DBE). All incubations, washes, and clearing were performed on a rotating wheel at room temperature, protected from light. Cleared samples were imaged in DBE, and stored in DBE at room temperature, protected from light.

**Imaging and image processing**. Whole-mount brightfield images of E12.5 and E13.5 embryos were acquired using a NSZ-405 Zoom Stereo Microscope at ×1.5 and ×4 magnifications. Images were adjusted for brightness and some images were reflected to have the tails oriented in the same direction for easy visualisation.

Fluorescent images of slide-mounted embryonic sections were imaged with a Zeiss AxioImager Z1 with an AxioCam HRm camera at a ×20 objective magnification. All images were processed using AxioVisionRel.4.8 and Fiji softwares. Sectioned embryonic images of *Gdf11*$^{-/-}$ with or without *Cdx2P:Hoxd12* tail regions were automatically tiled on the Zeiss AxioImager Z1 and compiled into one image using AxioVisionRel.4.8 in-built image stitching function MosaiX. Tiled image and individual image size ratios were maintained to ensure direct comparisons between tiled images. Single-cell layer images of E12.5 split tail tips were acquired using Zeiss AxioImager Z1 in Apotome mode (Gitter 7.5 LP/mm gef; 423662-0000-000) and ×20 objective magnification. Linear adjustments, such as for "Brightness" or "Contrast", in which the same change is made to each pixel according to a linear function, were used on whole immunofluorescence images using Fiji (ImageJ2.0) or AxioVision Software. Fluorescence images of individual serial sections were uniformly processed using identical linear adjustments. For the cropped regions of apotome acquired split tail regions linear adjustments were made to all images and background subtraction was applied to the single-fluorescence images to extract/distinguish the signal above autofluorescence.

Images of E18.5 skeletal preparations and E10.5 embryos following wholemount RNA in situ hybridisations were acquired with a Vision Dynamic BK Lab System at the Monash University Paleontology Lab. Images were taken with a Canon 5d MkII with a 100 mm Macro lens (focus stop 1:3/1:1). Multiple images were taken to extend the focal depth and stacked in ZereneStacker using the PMax algorithm. Scales were determined via known pixel counts for each focus stop in Adobe Photoshop.

For whole-mount immunofluorescence-stained embryos, Z sections were acquired every 2 µm on a lightsheet microscope (Ultramicroscope 2, LaVision, Miltenyi Biotec GMbH). Image stacks were imported into Imaris (v9.6; BitPlane, Oxford Instrument) and visualised in 3D. Contrast was adjusted for each channel and segmentation was performed using the Surface tool. Pictures and animations were created in Imaris.

Adobe Illustrator was used to compose all multipanel figures in this study.

**Statistics and reproducibility**. All skeletal phenotyping was performed independently by two investigators, blinded to genotype. No statistical method was used to predetermine sample size. For gene expression analyses, Ct values greater than the manufacturer recommended threshold were excluded.

**Reporting summary**. Further information on research design is available in the Nature Research Reporting Summary linked to this article.

## Data availability
The raw Fluidigm qPCR data generated in this study are provided in the Source data file. Summary tables of skeletal scoring data are available as supplementary tables 1–3. Supplementary Movies are available under the link: [https://figshare.com/s/78383c7beb9b85c959bf]. Source data are provided with this paper.

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

## Acknowledgements

We acknowledge the excellent technical assistance provided by Monash Gene Modification Platform, Monash Animal Research Platform, MHTP Medical Genomics Facility, Monash MicroImaging, Monash FlowCore, Andrew Langendam, Jeffrey Stilwell and Monique Centrone. We thank Irina Ruf for comments on the manuscript. The *Gdf11* mutant mouse line was kindly provided by Professor Se-Jin Lee. The goat anti-rabbit-AF790 was a kind gift of Dr. Jesse Di Cello. G.M.H. and Y.-C.C. are supported by an Australian Government Research Training Program Scholarship. J.M.P. is supported by an Australian Research Council Future Fellowship. This work was supported by National Health and Medical Research Council Project Grant APP1051792 to E.M. and Australian Research Council Discovery Project DP180102157 to E.M. and J.M.P. The Australian Regenerative Medicine Institute is supported by grants from the State Government of Victoria and the Australian Government.

## Author contributions

Conceptualisation, methodology and formal analysis, G.M.H., V.C.G. and E.M.; investigation, G.M.H., V.C.G., S.F.L.W., H.B., J.M., Y.C., C.M.N., J.F., J.C., F.J.R. and E.M.; writing—original draft, G.M.H. and E.M.; writing—review and editing, all authors; funding acquisition, E.M. and J.M.P.; supervision, F.J.R., J.M.P. and E.M.

## Competing interests

The authors declare no competing interests.
