## [Peer Review File · Nature Communications]

Breaking constraint of mammalian axial formulaeReviewers' Comments:

Reviewer #1:

Remarks to the Author:

This study that uses standard technique shows that the axial vertebral patterns can be changed upon modulating key factors like Hox genes, and describes how constraints of mammalian axial formulae can be broken by manipulating the activities of GDF11, retinoic acid, and microRNA-196 paralogs, all of which affect Hox gene expression. Also, utilizing iPSCs to generate in vitro model system and monitor gene expression patterns according to cell-state transitions in axial elongation is interesting. There are, however, a number of weaknesses in this paper.

First, data on GDF11, retinoic acid, and microRNA-196 paralogs are not necessarily new. The role of GDF11 in anterior/posterior patterning of the axial skeleton have been well documented in multiple papers including, but not limited to, MaPherron, et al. (Nat. Genet., 1999), Lee, et al. (Dev. Biol., 2010), Lee, et al. (PNAS, 2013), Suh, et al. (J. Cell. Physiol., 2019) etc. Especially, Lee, et al. has clearly shown that GDF11 signaling controls retinoic acid activity for vertebral development in their paper (Dev. Biol., 2010). Authors also used the same pan retinoic acid receptor antagonist, AGN193109, which was used and validated in Lee's paper (Dev. Biol., 2010), to show that the skeletal phenotypes can be affected by AGN193109. The role of microRNA-196 paralogs in regulation of vertebral number and identity was well documented by Wong, et al. (PNAS, 2015) as the authors acknowledge.

Second, authors generated transgenic mice expressing either Hoxd11 or Hoxd12 under the control of Cdx2 to restore Gdf11 mutant phenotype. However, caution must be taken to interpret the ectopic expression of Hox gene data since they may not recapitulate the endogenous functions of Hox genes. Third, authors describes that they found a surprising dose-dependent effect of GDF11 on axial formulae that has not been previously appreciated in the section titled as "Gdf11 and miR-196 synergistically constrain TVN". However, Lee, et al. has clearly shown that not only decreased activities of GDF11, but also increased activities of GDF11 in Gasp2^{-/-} and Fst^{-/-} background affect the axial vertebral patterning in a dose-dependent manner (PNAS, 2013).

Finally, the literature cited misses some important work on GDF11 such as the work by Lee, et al. (PNAS, 2013 Sep 24;110(39):E3713-22. Regulation of GDF-11 and myostatin activity by GASP-1 and GASP-2) and Suh, et al. (J. Cell. Physiol., 2019 Dec;234(12):23360-23368. Growth differentiation factor 11 locally controls anterior-posterior patterning of the axial skeleton).

Reviewer #2:

Remarks to the Author:

Here McGlinn and colleagues study the impact of a series of combinatorial effect of three important morphogenetic systems, Gdf11, miR-196 and RA, on the axial identity of mammalian vertebrae, through a series of genetic deletions combined with drug treatments of both iPSC-derived and mouse embryo models.

Through these elegant experiments, the authors find that these three signaling pathways finely control the expression of central and posterior Hox genes in order to constrain the positional identity of vertebrae and their number within each segment.

I have very little to add to such a complete piece of work but, if I'm allowed to, I would like to say that I have missed a deeper discussion about the putative importance of Hox temporal collinearity and its control by the systems they study here. If anything, it seems to me that the results described in this manuscript come to confirm the importance of the Hox clock in laying out the positional identity along the main body axis during the anterior-posterior elongation, as also elegantly discussed by Jacqueline Deschamps and Denis Duboule (<http://www.genesdev.org/cgi/doi/10.1101/gad.303123.117>). If the authors consider it appropriate, I recommend including a mention to the Hox clock in their discussion.

Very minor points:

-Fig. 1c, regarding the background genotypes, in the most-right panel, shouldn't it read miR-196a1^{-/-} instead of miR196a1^{+/-}? In other two panels I guess it should read a2^{-/-} instead of a2^{-/}?

-Fig. 3a, the genotypes are not centered in their corresponding rows.

Reviewer #3:

Remarks to the Author:

In this manuscript, Hauswirth and colleagues examine the molecular mechanisms controlling vertebral number in mice. Through an impressive, well-controlled, series of genetic and biochemical manipulations, the authors provide compelling evidence of the synergistic and independent roles of Gdf11, Mir196 and RA signalling, in the control of total vertebral number. Compound mutants in which all alleles of Mir196 and Gdf11 are absent display dramatic expansions in the number of thoracic and lumbar elements. The phenotypes presented offer a rare opportunity to examine the mechanisms underpinning this constraint in vivo along the length of AP axis.

The authors demonstrate that the constraint on vertebral number at different positions is associated with perturbations in the expression of Hox genes. Although these are well established regulators of the body plan, we know very little about the individual roles of Hox genes, in part due to their functional redundancy. Adding to recent observations in fish, the authors provide further evidence that posterior Hox genes can function in the control of segment number (not simply homeotic transformations). In contrast to other posterior Hox13 genes, Hoxd11/d12 do not profoundly truncate the axis, providing evidence for potentially divergent roles within the posterior cluster. It is not yet clear how Hoxd11/d12 operate in this context.

A minor comment, relates to the in vitro system. The addition of GDF11 in vitro clearly restrains an otherwise continual increase in trunk Hox genes in the TKO mutant cells. This seems a plausible explanation for the increase in vertebral number observed in vivo when both copies of Gdf11 are removed in the TKO background. Nonetheless, the trunk to tail transition can still occur, suggesting alternative signals eventually promote the transition. Can the authors comment on this, and how this may fit with recent studies eg Mouilleau et al 2021 (doi:10.1242/dev.194514) and similar in vivo observations suggesting that the transition from thoracic to lumbar Hox expression at least in vitro can be promoted by the combined action of FGF and GDF11.

Overall, the wealth of data presented in this manuscript are of a very high standard, are novel and valuable contributions, and support the conclusions. I commend the authors on their efforts.

Hauswirth, Garside et al.,

Response to Reviewer #1:

This study that uses standard technique shows that the axial vertebral patterns can be changed upon modulating key factors like Hox genes, and describes how constraints of mammalian axial formulae can be broken by manipulating the activities of GDF11, retinoic acid, and microRNA-196 paralogs, all of which affect Hox gene expression. Also, utilizing iPSCs to generate in vitro model system and monitor gene expression patterns according to cell-state transitions in axial elongation is interesting. There are, however, a number of weaknesses in this paper.

First, data on GDF11, retinoic acid, and microRNA-196 paralogs are not necessarily new. The role of GDF11 in anterior/posterior patterning of the axial skeleton have been well documented in multiple papers including, but not limited to, MaPherron, et al. (Nat. Genet., 1999), Lee, et al. (Dev. Biol., 2010), Lee, et al. (PNAS, 2013), Suh, et al. (J. Cell. Physiol., 2019) etc. Especially, Lee, et al. has clearly shown that GDF11 signaling controls retinoic acid activity for vertebral development in their paper (Dev. Biol., 2010). Authors also used the same pan retinoic acid receptor antagonist, AGN193109, which was used and validated in Lee's paper (Dev. Biol., 2010), to show that the skeletal phenotypes can be affected by AGN193109.

We agree and had referenced the work of Leet et al., 2010 in introducing our quadruple KO + AGN assessment:

Pg 7.

In this context, the partial rescue of *Gdf11*^{-/-} truncation by reducing endogenous levels of Retinoic acid (RA) signalling³² led us to examine the potential for more elaborate redundancy or synergy between the three signalling/regulatory pathways. (³² Lee et al., 2010).

The role of microRNA-196 paralogs in regulation of vertebral number and identity was well documented by Wong, et al. (PNAS, 2015) as the authors acknowledge. Second, authors generated transgenic mice expressing either *Hoxd11* or *Hoxd12* under the control of *Cdx2* to restore *Gdf11* mutant phenotype. However, caution must be taken to interpret the ectopic expression of Hox gene data since they may not recapitulate the endogenous functions of Hox genes.

We agree, and had stated this caveat in the discussion:

Pg 13.

This model does not preclude a role for high levels of posterior *Hox* expression in terminating axial elongation^{17,31,42}, and indeed our data on ectopic *Hoxd11/Hoxd12* can be viewed as supporting this. However, caution should be taken when interpreting overexpression studies⁴⁰ while awaiting the genetic deletion of *Hox12* and *Hox13* paralog groups in the mouse.

Third, authors describes that they found a surprising dose-dependent effect of GDF11 on axial formulae that has not been previously appreciated in the section titled as "Gdf11 and miR-196 synergistically constrain TVN". However, Lee, et al. has clearly shown that not only decreased activities of GDF11, but also increased activities of GDF11 in *Gasp2*^{-/-} and *Fst*^{-/-} background affect the axial vertebral patterning in a dose-dependent manner (PNAS, 2013).

We agree that an increase in 1 thoracic element in *Gdf11*^{+/-} embryos has been previously published as early as 1999, though whether this represented a homeotic transformation (ie an ultimate loss of an element in the caudal region) or a meristic increase was unknown (we show it is meristic). Moreover, the two additional tail vertebrae that form in *Gdf11*^{+/-} embryos has not previously been published. The surprising aspect that we refer to is that the *Gdf11*^{-/-} homozygous embryos have a truncated main body axis while the heterozygous embryos have an elongated one and have revised the sentence to clarify this point as follows:

Pg 6: a surprising dose-dependent effect of Gdf11 on TVN that has not previously been appreciated.

Finally, the literature cited misses some important work on GDF11 such as the work by Lee, et al. (PNAS, 2013 Sep 24;110(39):E3713-22. Regulation of GDF-11 and myostatin activity by GASP-1 and GASP-2) and Suh, et al. (J. Cell. Physiol., 2019 Dec;234(12):23360-23368. Growth differentiation factor 11 locally controls anterior-posterior patterning of the axial skeleton).

We apologise that we could not go into the full background details for every factor (Gdf11, miR-196 or RA). We have updated the manuscript to include reference to Suh et al., pg 4/17. The section on Gdf11 within the introduction was focused on factors that constrain vertebral number regionally – the Gdf11 loss-of-function data in Lee 2013 was the same as McPherron 1999 (cited) and so we have not included the Lee paper here. However we have included the work of McPherron 2009, pg8.

Response to Reviewer #2:

Here McGlinn and colleagues study the impact of a series of combinatorial effect of three important morphogenetic systems, Gdf11, miR-196 and RA, on the axial identity of mammalian vertebrae, through a series of genetic deletions combined with drug treatments of both iPSC-derived and mouse embryo models. Through these elegant experiments, the authors find that these three signaling pathways finely control the expression of central and posterior Hox genes in order to constrain the positional identity of vertebrae and their number within each segment.

I have very little to add to such a complete piece of work but, if I'm allowed to, I would like to say that I have missed a deeper discussion about the putative importance of Hox temporal collinearity and its control by the systems they study here. If anything, it seems to me that the results described in this manuscript come to confirm the importance of the Hox clock in laying out the positional identity along the main body axis during the anterior-posterior elongation, as also elegantly discussed by Jacqueline Deschamps and Denis Duboule (<http://www.genesdev.org/cgi/doi/10.1101/gad.303123.117>). If the authors consider it appropriate, I recommend including a mention to the Hox clock in their discussion.

We thank Reviewer 2 for their positive comments on our manuscript, and have included discussion of the Hox clock as follows:

Pg13

At a molecular level, the 5'-to-3' sequential activation of *Hox* cluster expression within axial progenitors over development time – the *Hox clock* (Deschamps and Duboule)- prefigures *Hox* spatial collinearity along the A-P axis (Forlani 2003; Neijts 2016), though the consequences of clock manipulation have not been easy to address. The mouse mutants presented here, all unified in their temporal control of global *Hox* transitions, offer a granular view as to the consequences of manipulating that *Hox* clock *in vivo*. We show that the speeding up, or the slowing down, of *Hox* cluster signatures result for the most part in serial vertebral transformations that associate with changes in total vertebral number. At a minimum, this implies a very tight association between *Hox* patterning and elongation mechanisms, though importantly, our data also revealed that posterior *Hox* genes have the capacity to positively influence axial elongation. This latter data, viewed along with similar results for trunk *Hox* genes in mouse³¹ and *Hox13* paralogs in zebrafish⁴⁰, allows us to propose a model whereby multiple post-occipital expressing *Hox* paralogs participate in construction of the main body axis, **not solely in its patterning**. We do not suggest that *Hox* genes are the primary drivers of axial elongation, but that a minimum level is required. This model does not preclude a role for high levels of posterior *Hox* expression in terminating axial elongation^{17,31,42}, and indeed our data on ectopic *Hoxd11/Hoxd12* can be viewed as supporting this. However, caution should be taken when interpreting overexpression studies⁴⁰ while awaiting the genetic

deletion of *Hox12* and *Hox13* paralog groups in the mouse. Why then has a role for *Hox* genes in shaping vertebral number not been apparent in the extensive *Hox* mouse mutant literature? Cumulative mutant analysis has shown that each individual vertebral element is patterned by at least two, if not more, *Hox* paralog groups³⁵, thus only by removal of multiple adjacent paralog groups would this function be revealed. Experimentally, this would be a complex undertaking and of questionable relevance from an evolutionary perspective. However, by altering the timing of collective *Hox* code transitions (i.e., *in vivo* pacing of the *Hox* clock) rather than altering *Hox* function *per se*, through changes in wide-spread post-transcriptional regulation (miR-196) or alterations in higher order signalling (Gdf11 haploinsufficiency), this unanticipated vertebrate *Hox* function has now been revealed.

Very minor points:

-Fig. 1c, regarding the background genotypes, in the most-right panel, shouldn't it read miR-196a1^{-/-} instead of miR196a1^{+/-}?

The genotype in Fig1c is correct: miR196a1^{+/-}; a2^{-/-};b^{-/-} (5 of the 6 alleles missing). In our earlier work, Wong et al., PNAS, 2015, we have shown that miR196a2^{-/-};b^{-/-} double KO embryos have identical axial skeleton changes to triple KOs. For this reason, the final embryos used for analysis include mice with at least 5 miR-196 alleles deleted – Please see Supp Table 1 where we clarify what animals were assessed, referred to as TKO*. The embryo chosen for the figure was the clearest representation of this phenotype – often these highly elongated skeletons are hard to image as the embryo curls towards the lower lumbar region and is no longer in clear focus. We would prefer to keep this image if possible.

In other two panels I guess it should read a2^{-/-} instead of a2^{-/}?

Thank you, you are correct. This has been corrected.

-Fig. 3a, the genotypes are not centered in their corresponding rows.

Thank you, you are correct. This has been corrected.

Response to Reviewer #3:

In this manuscript, Hauswirth and colleagues examine the molecular mechanisms controlling vertebral number in mice. Through an impressive, well-controlled, series of genetic and biochemical manipulations, the authors provide compelling evidence of the synergistic and independent roles of Gdf11, Mir196 and RA signalling, in the control of total vertebral number. Compound mutants in which all alleles of Mir196 and Gdf11 are absent display dramatic expansions in the number of thoracic and lumbar elements. The phenotypes presented offer a rare opportunity to examine the mechanisms underpinning this constraint *in vivo* along the length of AP axis.

The authors demonstrate that the constraint on vertebral number at different positions is associated with perturbations in the expression of *Hox* genes. Although these are well established regulators of the body plan, we know very little about the individual roles of *Hox* genes, in part due to their functional redundancy. Adding to recent observations in fish, the authors provide further evidence that posterior *Hox* genes can function in the control of segment number (not simply homeotic transformations). In contrast to other posterior *Hox13* genes, *Hoxd11/d12* do not profoundly truncate the axis, providing evidence for potentially divergent roles within the posterior cluster. It is not yet clear how *Hoxd11/d12* operate in this context.

A minor comment, relates to the *in vitro* system. The addition of GDF11 *in vitro* clearly restrains an otherwise continual increase in trunk *Hox* genes in the TKO mutant cells. This seems a plausible explanation for the increase in vertebral number observed *in vivo* when both copies

of Gdf11 are removed in the TKO background. Nonetheless, the trunk to tail transition can still occur, suggesting alternative signals eventually promote the transition. Can the authors comment on this, and how this may fit with recent studies eg Mouilleau et al 2021 (doi:10.1242/dev.194514) and similar *in vivo* observations suggesting that the transition from thoracic to lumbar Hox expression at least *in vitro* can be promoted by the combined action of FGF and GDF11.

Overall, the wealth of data presented in this manuscript are of a very high standard, are novel and valuable contributions, and support the conclusions. I commend the authors on their efforts.

Thank you also to Reviewer 3 for their positive comments on our manuscript.

The differentiation protocol we use maintains FGF2 across all days, with or without exogenous GDF11 exposure (please see Fig 1A). So at least *in vitro*, FGF2 alone is not capable of activating a posterior Hox code in the absence of Gdf11/miR-196, at the concentrations used. Whether this is different *in vivo* is hard to say, though it has been shown that Gdf8 acts redundantly with Gdf11 in inducing the T-to-T transition suggesting a second plausible candidate. It may be a mix of many signals, in the absence of Gdf11, that ensures the T-to-T eventuates, even if late.

We have added a sentence addressing this point:

Pg 8:

It is important to note that in these embryos, and all embryos assessed in this study, progression through the T-to-T transition always occurred, albeit late. This indicates that the regulatory synergism promoting this key transition was yet to be fully depleted, with prime candidates supporting the eventual T-to-T in compound mutant embryos being Gdf8 (McPherron et al., 2009) and potentially FGF signalling (Mouilleau et al., 2021).

Reviewers' Comments:

Reviewer #1:

Remarks to the Author:

Since the strain background differences of mice can affect the axial formulae, the strain backgrounds of Hoxd11 OE (Cdx2P:Hoxd11) and Hoxd12 OE (Cdx2P:Hoxd12) transgenic mice should be described.

Reviewer #2:

None

Reviewer #3:

Remarks to the Author:

The authors have addressed the reviewer comments. I recommend the work for publication.

RESPONSE TO REVIEWERS' COMMENTS

Reviewer #1 (Remarks to the Author):

Since the strain background differences of mice can affect the axial formulae, the strain backgrounds of Hoxd11 OE (Cdx2P:Hoxd11) and Hoxd12 OE (Cdx2P:Hoxd12) transgenic mice should be described.

We have included background information for these line on Pg. 18 as follows:

Cdx2P:Hoxd11 (*Hoxd11^{OE}*) and *Cdx2P:Hoxd12* (*Hoxd12^{OE}*) transgenic mice were generated in-house by pronuclear injection into C57BL/6J zygotes according to standard protocols.

Reviewer #3 (Remarks to the Author):

The authors have addressed the reviewer comments. I recommend the work for publication.